# Residential Exposure to Dampness Is Related to Reduced Level of Asthma Control among Adults

**DOI:** 10.3390/ijerph191811338

**Published:** 2022-09-09

**Authors:** Maritta S. Jaakkola, Henna Hyrkäs-Palmu, Jouni J. K. Jaakkola

**Affiliations:** 1Center for Environmental and Respiratory Health Research and Biocenter Oulu, Population Health, University of Oulu, FI-90014 Oulu, Finland; 2Finnish Meteorological Institute, FI-00101 Helsinki, Finland

**Keywords:** asthma control, bronchial asthma, indoor dampness, indoor visible mold, mold odor

## Abstract

We assessed potential relations between indicators of indoor dampness and mold exposures at home and the level of asthma control among adults. The present population-based cross-sectional study, the Northern Finnish Asthma Study (NoFAS), included 1995 adult subjects with bronchial asthma who replied to study questionnaires (response rate: 40.4%). The Asthma Control Test (ACT) was used as the measure of asthma control. We calculated the mean difference in ACT score (ACT_difference_) and the risk ratio (RR) of poor asthma control (ACT ≤ 19) for the exposure and reference groups and applied Poisson regression to adjust for potential confounding. Exposure to indoor dampness at home was related to a significantly reduced level of asthma control (ACT_difference_: −0.83, 95% CI: −1.60 to −0.07), especially among men (ACT_difference_: −2.68, 95% CI: −4.00 to −1.37). Water damage (aRR = 1.29, 95% CI: 1.01, 1.65) and indoor dampness, especially among men (aRR = 1.92, 95% CI: 1.15, 3.20), increased the risk of poor asthma control. We provide evidence that exposure to indoor dampness at home reduces asthma control among adults, especially in men. Indoor visible mold and mold odor were not significantly related to asthma control. Advice on how to prevent indoor dampness at home should be an important part of asthma management.

## 1. Introduction

There is increasing evidence that exposure to molds indoors increases the risk of developing bronchial asthma as well as the occurrence of asthma-related symptoms among adult subjects who have diagnosed asthma [1,2,3]. In 2004, the Asthma Control Test (ACT) was developed as a standardized, questionnaire-based method to assess the level of the control of asthma and is easily applicable to the management of bronchial asthma [4], for example, in the outpatient clinic, where it may not be possible to carry out more sophisticated lung function tests or to measure markers of inflammation. There is some previous epidemiological evidence on the determinants of asthma control measured by applying the ACT score. Abrahamsen and colleagues characterized the risk factors for poor asthma control in a cross-sectional study of the general population of Telemark County, Norway [5]. They used an ACT score ≤ 19 as the measure of poor asthma control. Female sex, low level of education, past and current smoking, obesity (BMI > 30), and occupational exposure to vapor, gas, dust, or fumes were reported as risk factors for poor asthma control, while age and the presence of allergy did not increase the risk of poor asthma control in the Norwegian study. Fernandez and colleagues [6] studied the relation between air pollution levels and the ACT score in Oviedo, Spain. In a multivariate linear regression analysis, NO_2_ levels predicted the ACT score, while SO_2_ and PM_10_ were not related to this outcome. Dumas and colleagues [7] reported that household exposure to disinfectants and cleaning products increased the risk of poor asthma control, measured as ACT ≤ 19, among elderly French women. Dumas and colleagues [8] also reported an association between the frequency of hand and arm hygiene tasks and poor asthma control, defined as an ACT score ≤ 19, among US nurses. The weekly use of sprays and chemicals was especially associated with poorly controlled asthma. 

In our systematic literature search, we did not identify any previous study that had addressed residential dampness and mold problems as determinants of poor asthma control. However, there is substantial epidemiological evidence showing that indoor dampness and mold problems increase the risk of asthma [1,2,3]. Furthermore, there is accumulating evidence that some environmental exposures, such as ambient air pollution and chemical exposures from cleaning agents, increase the risk of poor asthma control [5,6,7,8]. Based on the literature review, we hypothesized that exposure to microbial agents or chemicals from damp structures in residential buildings influences the level of asthma control. We tested this hypothesis by elaborating potential relations between indicators of indoor dampness and mold exposures at home and the level of asthma control among adult subjects with physician-diagnosed asthma.

## 2. Methods

### 2.1. Study Design and Study Population

This study was based on the population-based cross-sectional Northern Finnish Asthma Study (NoFAS). A total of 1995 adult subjects (response rate: 40.4%) 17–73 years old who had physician-diagnosed bronchial asthma based on their receipt of the special reimbursement right for asthma medications from the Finnish National Social Insurance Institution (NSII) and who were living in Northern Finland answered the NoFAS questionnaire and the St George’s Respiratory Questionnaire [9].

### 2.2. Exposure Assessment

Exposure was assessed based on the study subject reporting exposure in the questionnaire. We applied four indicators of exposure, which were defined based on the answers to the following structured questions: (1) Water damage: “Have you ever had a relatively great water damage in your dwelling? (for example, leakage of a water pipe or roof)” (yes, during the past 12 months; yes, 1–3 years ago; yes, only earlier; no). (2) Dampness: “Have you ever noticed wet spots or changes in the color on the ceiling, floor or walls of your dwelling?” (yes, during the past 12 months; yes, 1–3 years ago; yes, only earlier; no). (3) Visible mold: “Have you ever noticed visible mold in your dwelling?” (yes, during the past 12 months; yes, 1–3 years ago; yes, only earlier; no). (4) Mold odor: “Have you smelled mold odor or stuffiness (such as in a cellar) in your dwelling during the past 12 months?” (yes, during the past 12 months; yes, 1–3 years ago; yes, only earlier; no). In addition, we formed an any-exposure indicator based on the presence of any of these four exposure indicators. The reference category comprised study subjects who answered “no” to all questions on potential exposures.

### 2.3. Health Outcomes

The outcome of interest was the level of asthma control, which was assessed based on replies to the questions of the Asthma Control Test (ACT) [4]. The ACT includes 5 questions: (1) “During the past 4 weeks, how much of the time did your asthma keep you from achieving much at work, school or at home?” (all of the time; most of the time; some of the time; a little of the time; none of the time), (2) During the past 4 weeks, how often did you experience shortness of breath? (more than once a day; once a day; 3 to 6 times a week; once or twice a week; not at all), (3) “During the past 4 weeks, how often did your asthma symptoms (including wheezing, coughing, shortness of breath, chest tightness and/or chest pain) wake you up at night or earlier than usually in the morning?” (4 or more nights a week; 2 or 3 nights a week; once a week; once or twice; not at all), (4) “During the past 4 weeks, how often have you needed your rescue inhaler or nebulizer containing asthma medication (such as salbutamol)?” (3 or more times per day; 1 or 2 times per day; 2 or 3 times per week; once a week or less frequently; not at all), and (5) “How would you rate your asthma control during the past 4 weeks?” (not at all controlled; poorly controlled; somewhat controlled; well controlled; completely or very well controlled). For every question, the first choice provided 1 point, and the last choice was assigned 5 points, so the ACT could have scores from 5 to 25. An ACT score of 5 indicated uncontrolled asthma, and 25 indicated totally or very well-controlled asthma. We also applied a dichotomous outcome variable, which indicated poor asthma control when the ACT score was < 19.

### 2.4. Statistical Methods

We estimated the relations between the presence of dampness or mold exposures at home and the ACT score. First, we compared the mean ACT score in different exposure categories. Second, we applied Poisson regression with the unity link function to estimate the adjusted ACT score differences between exposed asthmatics and unexposed asthmatics by applying the formula:ACT_difference_ = ACT_exposed_ − ACT_reference_

The relations between the presence of exposures at home and the risk of poor asthma control score were analyzed by Poisson regression using a logarithmic link function, and the results are presented as risk ratios (RRs) as measures of effect. For adjusted RRs, adjustment was made for the following potential confounders: gender, age, body mass index (BMI), smoking, secondhand smoke exposure, education, and cohabitation. In the full model, adjustment was also made for some concurrent diseases, including rhinitis, chronic obstructive pulmonary disease (COPD), and cardiovascular diseases. Multivariate analyses were carried out by applying the GENMOD procedure in the SAS software (SAS 9.4, SAS Institute, Inc., Cary, North Carolina).

## 3. Results

### 3.1. Characteristics of the Study Population

Table 1 shows the distribution of some relevant characteristics of the study population according to their exposure status. The study population included 1995 subjects. The exposure information was missing for 18 participants (0.9%), and the analyses were conducted among 1977 participants. A total of 448 participants (22.66%) reported at least one of the exposures at home. Those reporting exposure to residential dampness and/or molds were somewhat more often women, significantly more often had high BMI, and were significantly more often smokers. 

### 3.2. Asthma Control in Relation to Indoor Dampness and Mold Exposures

Table 2 shows the effect of different residential dampness and mold exposures on the ACT level. Exposure to indoor dampness at home was related to a significantly reduced ACT score, especially among men, with their ACT_difference_ being −2.68 (95% CI −4.00 to −1.37) compared to the reference category with no exposure. The adverse effect of residential dampness was also detected in the total population, with the ACT level being −0.83 (−1.60 to −0.07). The mean ACT score was not statistically significantly related to the other exposure indicators.

Table 3 presents the crude and adjusted risk ratios (RRs) for the effects of residential dampness and mold exposures on the risk of poor asthma control. The risk of poor asthma control was related to both indoor water damage and dampness. In Poisson regression with a logarithmic link function, the adjusted RR related to water damage was 1.29 (95% CI 1.01, 1.65) among the total population, and the effect estimates were similar for both men and women. The effect estimate for indoor dampness was elevated among the total study population (adjusted RR = 1.27, 95% CI 0.97, 1.66) but stronger and statistically significant among men (adjusted RR = 1.92, 95% CI 1.15, 3.20). The RR for residential dampness was not much elevated among women (adjusted RR = 1.05, 95% CI 0.76, 1.43).

## 4. Discussion

We conducted a population-based cross-sectional study of 1995 adult subjects who had asthma based on their receipt of the special reimbursement right for asthma medications from the National Social Insurance Institution of Finland. This means that their asthma had been diagnosed following the national asthma guidelines: i.e., their diagnosis fulfilled the criteria required by the NSII before it warrants the special reimbursement right. Our results showed that exposure to indoor dampness was related to a significantly reduced ACT score among men, which is in line with the results of a previous meta-analysis that showed a significantly increased risk ratio of incident asthma (effect estimate (EE) 1.33, 95% CI 1.12 to 1.56) in relation to self-reported exposure to indoor dampness [2]. The risk of poor asthma control was related to water damage in men and women combined and to indoor dampness in men. The present study did not detect a significantly increased risk related to indoor visible mold or mold odor, which may be explained by the fact that indoor dampness problems detected at home are most likely repaired rapidly before significant mold growth has time to develop.

### 4.1. Validity of Results

The response rate in this study was 40%, which was satisfactory but somewhat lower than in our previous epidemiological studies [1,10]. Because the cases of bronchial asthma were identified through the asthma medication reimbursement files of the NSII, the invitation letter and questionnaires were sent by the NSII. To guarantee anonymity of the study subjects, Finnish law does not allow investigators to send any letters directly to potential participants identified through the NSII. This is likely to have contributed to this lower response rate, as the investigators were not allowed to send any reminder letters to potential participants. However, the sample size was large, with 1995 participants, and the study population covered a wide adult age range (18 to 71 years) to get a good picture of how indoor dampness and mold problems potentially influence asthma control in Northern Finland.

Exposure was assessed based on replies to the NoFAS questionnaire inquiring about the potential presence of indoor water damage, damp stains, and/or other marks of structural dampness, visible mold, and/or mold odor at home in the past 12 months. These four indicators of exposure have been used in several previous epidemiological studies, which have been summarized in two systematic reviews and meta-analyses [2,3]. They were based on occupants’ own observations at home. There is empirical evidence that reported visible mold growth and signs of dampness correlate well with increased numbers of mold spores and mold species measured in the air [11,12,13,14]. The correlation between visible inspection and measurement of high indoor mold levels was found to be accurate in 80 percent of cases; in the remaining 20% of cases, high concentrations of mold were not reflected in visual inspection, which suggested that they were from mold growth in hidden areas, for example, behind walls or under carpets [11]. Assessments by both occupants and inspectors have strengths and weaknesses. Occupants may not notice all of the exposure details, but their observation period covers the whole time period of relevance for health effects. Trained inspectors can make very accurate observations about the environmental conditions, but their observations may not be valid for a longer time period, which may be more relevant for the effects on asthma control. An important validity issue in health effect assessment is whether there is any systematic measurement error; i.e., do occupants with health problems observe the environmental conditions similarly or differently compared to those who are healthy?

The outcome of interest was the level of asthma control, which was assessed based on replies to a self-administered questionnaire that included five previously validated, standardized ACT questions [4]. Several previous epidemiological studies have assessed risk factors for asthma control using an ACT score ≤ 19 as the measure of poor asthma control. We compared the mean ACT score between the exposed and reference groups. The use of the mean ACT score provides more information on asthma control than the dichotomous ACT variable, as the continuous score has more power to detect differences between the groups. Furthermore, we also applied the dichotomous outcome for poor asthma control as an additional health outcome.

The multivariate analyses applied Generalized Linear Models with Poisson distribution and identity link and logarithmic link functions. Adjustment was made for age, BMI, personal smoking, exposure to secondhand smoke, and education. These covariates have been reported to be risk factors for poor asthma control in previous studies [5]. We did not adjust for air pollution or occupational exposures, which have also been reported as risk factors for poor asthma control [6,7,8]. We also conducted sensitivity analyses using poor asthma control (ACT ≤ 19) as a dichotomous health outcome. 

### 4.2. Synthesis with Previous Knowledge

Based on a systematic literature search, the present study is the first epidemiological study on the relations between residential dampness and mold problems and poor asthma control. Concerning other environmental determinants of asthma control, there is evidence that exposure to ambient air pollution [6], as well as occupational exposures [7,8], such as disinfectants and cleaning agents, affects the level of asthma control. Unfortunately, we did not have enough data on these two categories of environmental exposures in the present study.

Previous toxicological studies have suggested biological mechanisms that could plausibly underly the presently detected adverse effect of indoor dampness on asthma control. These include inflammatory, immunosuppressant, and even cytotoxic responses in the airways as responses to different dampness-related exposures [2,3]. Concerning the development of asthma, the following causal pathway has been proposed in the literature [2]:Water damage => Dampness => Visible molds => Mold odor => Onset of asthma

As in the present study, water damage and especially indoor dampness were identified as the important determinants for reduced ACT scores. Thus, we hypothesize that detecting water damage or indoor dampness at home leads to preventive measures and the resolution of dampness problems at an early stage, which explains why indoor molds were not identified as determinants of asthma control in the present study.

## 5. Conclusions

The present study provides new evidence that exposure to water damage and indoor dampness at home reduces the level of asthma control among adults with diagnosed asthma. Men were found to be especially susceptible to this effect on reduced asthma control. Advice on how to prevent indoor dampness at home should be included as an important part of asthma management for adults. Moreover, any water damage and/or indoor dampness detected should be repaired rapidly when such exposures are detected. 

## Figures and Tables

**Table 1 ijerph-19-11338-t001:** Characteristics of the study population included in the analyses (n = 1977). The Northern Finnish Asthma Study (NoFAS).

Characteristic	No Exposure N (%)	Exposure N (%)	Total N (%)	*p*-Value
**Total study** **population**	1529 (77.34)	448 (22.66)	1977 (100.00)	
**Age**				0.3173
<30	173 (11.31)	38 (8.48)	211 (10.67)	
30–39	210 (13.73)	57 (12.72)	267 (13.51)	
40–49	263 (17.20)	90 (20.09)	353 (17.86)	
50–59	494 (32.31)	151 (33.71)	645 (32.63)	
>60	389 (25.44)	112 (25.00)	501 (25.34)	
**Gender**				0.3217
Men	537 (35.12)	146 (32.59)	683 (34.55)	
Women	992 (64.88)	302 (67.41)	1294 (65.45)	
**Body mass index ***				0.0045
<20	60 (4.00)	22 (5.00)	82 (4.23)	
20–25	492 (32.80)	139 (31.59)	631 (32.53)	
25–30	565 (37.67)	131 (29.77)	696 (35.88)	
30–35	265 (17.67)	101 (22.95)	366 (18.87)	
>35	118 (7.87)	47 (10.68)	165 (8.51)	
**Education ***				0.3623
Comprehensive or upper secondary	394 (25.85)	100 (22.52)	494 (25.10)	
Vocational school	777 (50.98)	237 (53.38)	1014 (51.52)	
Academic degree	353 (23.16)	107 (24.10)	460 (23.37)	
**Smoking ***				0.0158
Never smoker	797 (52.50)	213 (48.19)	1010 (51.53)	
Ex-smoker	462 (30.43)	127 (28.73)	589 (30.05)	
Current smoker	259 (17.06)	102 (23.08)	361 (18.42)	
**Second-hand smoke ***				0.2155
Yes	972 (64.07)	298 (67.27)	1270 (64.80)	
No	545 (35.93)	145 (32.73)	690 (35.20)	

* Information missing: BMI (n = 37), education (n = 9), smoking (n = 17), second-hand smoke (n = 17).

**Table 2 ijerph-19-11338-t002:** Residential dampness and mold exposures and the adjusted difference in the ACT score (95% confidence interval) in comparison to no indoor dampness or mold exposure at home. The Northern Finnish Asthma Study 2012.

Study Population	No Exposure (Reference) Mean ACT	Water Damage ACT_difference_ (95% CI)	Indoor Dampness at Home ACT_difference_ (95% CI)	Visible Mold ACT_difference_ (95% CI)	Mold Odor ACT_difference_ (95% CI)	Any Exposure Indoors at Home ACT_difference_ (95% CI)
Total population	20.58	−0.19 (−0.97 to 0.59)	−0.83 * (−1.60 to −0.07)	−0.46 (−1.42 to 0.51)	0.01 (−0.77 to 0.78)	−0.07 (−0.51 to 0.37)
Men	20.54	−0.18 (−1.63 to 1.27)	−2.68 * (−4.00 to −1.37)	−0.94 (−2.74 to 0.87)	−0.02 (−1.49 to 1.44)	−0.59 (−1.42 to 0.23)
Women	20.61	−0.19 (−1.12 to 0.74)	0.06 (−0.88 to 1.00)	−0.35 (−1.49 to 0.78)	0.00 (−0.92 to 0.92)	0.04 (−0.59 to 0.57)

* The effect on ACT was statistically significant (*p* < 0.05).

**Table 3 ijerph-19-11338-t003:** Residential dampness and mold exposures and the risk of poor asthma control. Adjusted risk ratios (RRs) (95% confidence interval) contrasted between exposure categories and the reference category of no exposure from Poisson regression analysis. The Northern Finnish Asthma Study 2012.

Study Population	No Exposure	Water Damage	Indoor Dampness	Visible Mold	Mold Odor	Any Exposure
	Reference	RR (95% CI)	RR (95% CI)	RR (95% CI)	RR (95% CI)	RR (95% CI)
**Crude**	1.00	1.30 (1.01, 1.67)	1.33 (1.01, 1.74)	1.07 (0.82, 1.41)	0.92 (0.72, 1.18)	1.14 (0.98, 1.31)
**Adjusted ***	1.00	1.29 (1.01, 1.65)	1.27 (0.97, 1.66)	1.03 (0.79, 1.34)	0.89 (0.70, 1.14)	1.06 (0.92, 1.22)
**Men**						
**Crude**	1.00	1.26 (0.82, 1.92)	2.12 (1.27, 3.52)	1.12 (0.71, 1.78)	0.94 (0.61, 1.43)	1.18 (0.92, 1.50)
**Adjusted ***	1.00	1.28 (0.85, 1.94)	1.92 (1.15, 3.20)	1.18 (0.76, 1.84)	0.89 (0.59, 1.35)	1.08 (0.85, 1.38)
**Women**						
**Crude**	1.00	1.32 (0.97, 1.80)	1.06 (0.77, 1.46)	1.05 (0.75, 1.48)	0.92 (0.68, 1.25)	1.11 (0.93, 1.33)
**Adjusted ***	1.00	1.29 (0.95, 1.75)	1.05 (0.76, 1.43)	0.99 (0.72, 1.37)	0.90 (0.67, 1.21)	1.04 (0.88, 1.24)

* Poisson regression using logarithmic link function, adjusted for gender, age, body mass index (BMI), smoking, secondhand smoke exposure, education, and cohabitation.

## Data Availability

Data will not be shared for reasons of confidentiality of the questionnaire data.

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
