# Peer review of "Residential Exposure to Dampness Is Related to Reduced Level of Asthma Control among Adults"

_ijerph, 2022, doi:10.3390/ijerph191811338_

Round 1

Reviewer 1 Report

The article represents clear result obtained while analysing a relationship between exposure factors and effectiveness of asthma control.

However, despite of its brevity, this paper still needs revision.|
First, I will suggest to change the article title. As the main outcome of the paper is the connection of the dampness and asthma control, expression “and Mold Problems” seems to be both extra and do not adding any sense. Therefore, the modified title could be as follows: “Residential Dampness affects Efficiency of Asthma Control in Adults” or so (it’s logical that adults without asthma do not need asthma control at all).

Other remarks are as follows:

Lines 21-22, you wrote: We provide evidence that exposure to indoor dampness at home reduces asthma control among adults with diagnosed asthma. However, indoor dampness can rather reduce effectiveness of asthma control, but not asthma control measures by itself, as it is follows from you original sentence. So, this expression should be reconsidered at this point and hereafter.

In addition, it is better to report about the obtained result in past tenses, so, the tenses utility in the Manuscript also should be checked. E.g. it is better to write: “We provided evidence that exposure …”

Lines 37-38, you wrote: “…potential relations between indicators of indoor dampness and mold exposures at home and asthma control among adult subjects with physician-diagnosed asthma”. It is better to clarify here again that efficiency, level, performance, success or so of asthma control but not the asthma control by itself is considered.

The same for line 64.

English in lines 60-61 and 65 needs revision.
For example, lines 60-61, which write “The reference category comprised no to all exposure indi- cators” could be refresaed as The reference category comprised no of any exposure indicators.

Line 65 writes: i) disadvantage related to, as well…. But disadvantage related to what? To asthma?
Line 67-68 writes: v) self- 67 assessment of asthma control during the past 4 weeks. – Asthma control measures, efficiency, scores or so – should be clarified.

Lines 72 and 73, better to write ACT scores
Table 1
The line “Missing” and numbers in it is completely unclear. I could not find a clear explanation in the text.

Line 114 writes: “Our finding that exposure to indoor dampness is..”” Should be Our findings suggested that… - check, please.

Line 155 – probably, ACT scores are mentioned

Lines 133-137 should be replaced into Method section

Lines 139-141: The same, it is method and repretitive with lines 79-80

Line 147: should be “dampness-related” – check

Line 148: should be “proposed based on the literature” – check

Line 153: probably, ACT scores are mentioned here again

Line 158, you wrote: “reduces asthma control among adults…”, it is better to say: reduces asthma control efficiency among adults…..”

Lines 160-161 write: “be included as an important part of asthma management for adults who have bronchial asthma”, it is better to write: “be included as an important part of asthma management for adults” as asthma management is not prescribed to people without asthma. It is enough to indicate in the line 158 that you mentioned adults with diagnosed asthma.

Author Response

Reviewer 1

Comments and Suggestions for Authors

COMMENT 1: The article represents clear result obtained while analysing a relationship between exposure factors and effectiveness of asthma control.

RESPONSE 1: We thank the reviewer for this encouraging assessment.

COMMENT 2: However, despite of its brevity, this paper still needs revision.

RESPONSE 2: We have now revised the manuscript, expanded it substantially and provide here the Point-by-point response to the Reviewers’ comments.

COMMENT 3: First, I will suggest to change the article title. As the main outcome of the paper is the connection of the dampness and asthma control, expression “and Mold Problems” seems to be both extra and do not adding any sense. Therefore, the modified title could be as follows: “Residential Dampness affects Efficiency of Asthma Control in Adults” or so (it’s logical that adults without asthma do not need asthma control at all)

RESPONSE 3: We thank the Reviewer for this comment. We have now modified the title into ‘Residential Exposure to Dampness is related to Reduced Level of Asthma Control among Adults’.

Other remarks are as follows:

COMMENT 4: Lines 21-22, you wrote: We provide evidence that exposure to indoor dampness at home reduces asthma control among adults with diagnosed asthma. However, indoor dampness can rather reduce effectiveness of asthma control, but not asthma control measures by itself, as it is follows from you original sentence. So, this expression should be reconsidered at this point and hereafter.

In addition, it is better to report about the obtained result in past tenses, so, the tenses utility in the Manuscript also should be checked. E.g. it is better to write: “We provided evidence that exposure …”

RESPONSE 4: We have modified this sentence into: ‘We provided evidence that exposure to indoor dampness at home reduces asthma control among adults, especially in men.’. P. 1, lines 19-20.

COMMENT 5: Lines 37-38, you wrote: “…potential relations between indicators of indoor dampness and mold exposures at home and asthma control among adult subjects with physician-diagnosed asthma”. It is better to clarify here again that efficiency, level, performance, success or so of asthma control but not the asthma control by itself is considered.

RESPONSE 5: We have now modified this sentence into: “We tested this hypothesis by elaborating potential relations between indicators of indoor dampness and mold exposures at home and the level of asthma control among adult subjects with physician-diagnosed asthma”. P.2, lines 56-59.

COMMENT 6: The same for line 64.

RESPONSE 6: We have corrected this in the same way as in Response 5. P.2, line 86.

 COMMENT 7: English in lines 60-61 and 65 needs revision.

For example, lines 60-61, which write “The reference category comprised no to all exposure indi- cators” could be refresaed as The reference category comprised no of any exposure indicators.

RESPONSE 7: We have now corrected this into: “The reference category comprised study subjects answering “no” to all questions on the exposure indicators.” P.2, lines 81-82.

COMMENT 8: Line 65 writes: i) disadvantage related to, as well…. But disadvantage related to what? To asthma?

RESPONSE 8: We have corrected this by presenting all ACT questions and we also give information on how the ACT scores were calculated. P. 2-3, lines 85-100.

COMMENT 9: Line 67-68 writes: v) self- 67 assessment of asthma control during the past 4 weeks. – Asthma control measures, efficiency, scores or so – should be clarified.

RESPONSE 9: We have clarified that this means the level of asthma control during the past 4 weeks. P.2-3, line 85-100.

COMMENT 10: Lines 72 and 73, better to write ACT scores

RESPONSE 10: We have corrected this into ‘…the level of ACT’. P.2, line 111.

COMMENT 11: Table 1

The line “Missing” and numbers in it is completely unclear. I could not find a clear explanation in the text.

RESPONSE 11: ‘Missing’ means here that the answer to that specific question was missing. First of all, exposure variables were missing for 18 subjects out of 1995, leaving 1977 subjects for the analyses. If there was missing information on the other variables applied in the analysis concerning the results presented, this led to the study subject being excluded from that analysis. In the revised version, the number of missing information is given in the footnote.

COMMENT 12: Line 114 writes: “Our finding that exposure to indoor dampness is”” Should be Our findings suggested that… - check, please.

RESPONSE 12: We have modified this sentence into: ‘Our results showed that exposure to indoor dampness is related to a significantly reduced ACT score among men, which is in line with the results of a previous meta-analysis that showed a significantly increased risk ratio of incident asthma (EE 1.33, 95% CI 1.12 to 1.56) in relation to self-reported exposure to indoor dampness [2].’  P.5, lines 173-175.

COMMENTS 13-16:

COMMENT 13: Line 155 – probably, ACT scores are mentioned

REPLY 13: We have corrected this sentence into: Thus, as in this study indoor dampness was identified as the important determinant of reduced ACT score, we hypothesize that detecting water damage or indoor dampness at home leads to preventive measures and repairment of dampness problems at an early stage, which explains why indoor molds were not identified as determinants of asthma control in this study. P.5, line 242-246.

COMMENT 14: Lines 133-137 should be replaced into Method section

REPLY 14: We follow the tradition to present in the beginning of Discussion the most important features of the study and the most important results shortly, so we would prefer to keep this.

COMMENT 15: Lines 139-141: The same, it is method and repretitive with lines 79-80

REPLY 15: See reply 14. We follow the tradition to discuss in the Validity of Results section of Discussiom all the major aspects of methodology of an epidemiological study at least shortly: i) Response rate, ii) Exposure assessment, iii) Outcome assessment, and iv) Control for confounding.

COMMENT 16: Line 147: should be “dampness-related” – check

REPLY 16: We have corrected this. P. 5, line 247.

COMMENT 17: Outcome Line 148: should be “proposed based on the literature” – check

REPLY 17: We have corrected this.

COMMENT 18: Line 153: probably, ACT scores are mentioned here again

REPLY 18: We have corrected this into ACT level.

COMMENT 19: Line 158, you wrote: “reduces asthma control among adults…”, it is better to say: reduces asthma control efficiency among adults…..”

COMMENT 19: We have corrected this into ACT level.

COMMENT 20: Lines 160-161 write: “be included as an important part of asthma management for adults who have bronchial asthma”, it is better to write: “be included as an important part of asthma management for adults” as asthma management is not prescribed to people without asthma. It is enough to indicate in the line 158 that you mentioned adults with diagnosed asthma.

REPLY 20: We have corrected this. P.5, line 252.

Reviewer 2 Report

Overall, the results presented in ijerph-1861674 could provide new insights for elucidating the triggers of asthma that are related to various allergens and improper indoor conditions.

While the results are interesting, the structure and content of the manuscript should be improved because the introduction is too short (only 8 lines - L30-38) with 4 references from which 2 are written by the first author (furthermore, the manuscript comprises only 6 references), the methodology is briefly presented and discussion does not compare the results with international findings presented in the literature.

For these reasons, the manuscript might be considered as “Communication” and not a full article as it is stated. However, even for a short communication, more required information should be included.

Abstract: the way of presenting the information is uncommon (use the required structure for an abstract and do not use numbers 1), 2), etc. Please read the instructions provided at https://www.mdpi.com/journal/ijerph/instructions

Introduction: The introduction serves the purpose of leading the reader from a general subject area to a particular field of research. It establishes the context of the research being conducted by summarizing current understanding and background information about the topic (using updated meaningful references), stating the purpose of the work in the form of the hypothesis, question, or research problem, and briefly explaining your rationale, methodological approach, highlighting the potential outcomes your study can reveal, and describing the remaining structure of the paper. The rationale should be described more clearly. Sick building syndrome should be described and associated with dampness/molds.

Discussion: This section is often considered the most important part of a research paper. The discussion will always connect to the introduction by way of the research questions or hypotheses you posed and the literature you reviewed, but it does not simply repeat or rearrange the introduction; the discussion should always explain how your study has moved the reader's understanding of the research problem forward from where you left them at the end of the introduction. Include limitations of the approach and future work to overcome the drawbacks.

On the other hand, other issues should be addressed after including the missing elements:

Important! The authors should clarify more the mechanisms and triggers related to the exacerbation of asthma. The presence of various allergens (e.g., house dust, molds feathers, hair, fur, etc.) and chemical emissions (from furniture, carpets, repellents, scents, wall paintings, etc.) in indoor environments has a major effect acting as asthma triggers. Outdoor exposure is influenced by air pollution, and pollen, having a significant seasonality. Genetic aspects are also important for the severity of asthma stages. Long-term controller medication effects should be also discussed. These were not considered as confounders and this requires explanations and assessment of the potential effect on the presented results.

An asthma attack is the result of an exacerbation of the oxidative stress from reactive oxygen species (ROS), manifested more frequently during the night or early morning, by the appearance of paroxysms with specific symptoms. Aggravating effects such as inflammation, hyperactivity and obstruction of the respiratory tract should be discussed.

Not lastly, table 1, which is a key element of the presented research lacks important information in the caption (values in the parentheses are percentages?). No statistical indicator was presented. Neither significance. ORs or RRs should be considered. Statistical power was not presented.

L150 presents a simplified causal pathway, dampness and associated molds being a component of the asthma triggers. This should be discussed according to the elements presented before.

Limitations should be discussed. The approach was solely based on questionnaires. No monitoring and measurements were performed.

English language should be checked by reformulating ambiguous statements and correcting typos (e.g. L 148 “was prosed”).

Author Response

Reviewer 2

COMMENT 2.1: Overall, the results presented in ijerph-1861674 could provide new insights for elucidating the triggers of asthma that are related to various allergens and improper indoor conditions.

REPLY 2.1: We thank for this encouraging comment.

COMMENT 2.2: While the results are interesting, the structure and content of the manuscript should be improved because the introduction is too short (only 8 lines - L30-38) with 4 references from which 2 are written by the first author (furthermore, the manuscript comprises only 6 references), the methodology is briefly presented and discussion does not compare the results with international findings presented in the literature.

For these reasons, the manuscript might be considered as “Communication” and not a full article as it is stated. However, even for a short communication, more required information should be included.

REPLY 2.2: We expanded the manuscript substantially, conducted new analyses and included 8 new references which cover risk factors for ACT. However, ours is the first study on indoor dampness and mold problems and asthma control.

COMMENT 2.3: Abstract: the way of presenting the information is uncommon (use the required structure for an abstract and do not use numbers 1), 2), etc. Please read the instructions provided at https://www.mdpi.com/journal/ijerph/instructions

REPLY 2.3: We have rewritten and shortened the abstract according to the instruction.

COMMENT 2.4: Introduction: The introduction serves the purpose of leading the reader from a general subject area to a particular field of research. It establishes the context of the research being conducted by summarizing current understanding and background information about the topic (using updated meaningful references), stating the purpose of the work in the form of the hypothesis, question, or research problem, and briefly explaining your rationale, methodological approach, highlighting the potential outcomes your study can reveal, and describing the remaining structure of the paper. The rationale should be described more clearly. Sick building syndrome should be described and associated with dampness/molds.

REPLY 2.4: We have now modified the Introduction applying Instructions by the Journal and expanded it from 8 to 25 lines. However, we did not add information on SBS, because the outcome of interest in this manuscript is the level of asthma control.

COMMENT 2.5: Discussion: This section is often considered the most important part of a research paper. The discussion will always connect to the introduction by way of the research questions or hypotheses you posed and the literature you reviewed, but it does not simply repeat or rearrange the introduction; the discussion should always explain how your study has moved the reader's understanding of the research problem forward from where you left them at the end of the introduction. Include limitations of the approach and future work to overcome the drawbacks.

REPLY 2.5: We have now revised the Discussion taking these comments into consideration.

COMMENT 2.6: Important! The authors should clarify more the mechanisms and triggers related to the exacerbation of asthma. The presence of various allergens (e.g., house dust, molds feathers, hair, fur, etc.) and chemical emissions (from furniture, carpets, repellents, scents, wall paintings, etc.) in indoor environments has a major effect acting as asthma triggers. Outdoor exposure is influenced by air pollution, and pollen, having a significant seasonality. Genetic aspects are also important for the severity of asthma stages. Long-term controller medication effects should be also discussed. These were not considered as confounders and this requires explanations and assessment of the potential effect on the presented results.

REPLY 2.6: We have now revised the Discussion on determinants of asthma control. However, asthma control is not the same as acute exacerbations of asthma symptoms and it is slightly different from long-term severity of asthma. We have also conducted analyses using a dichotomous “poor asthma control” as a secondary health outcome. Potential confounding is also discussed on P. 6, lines 220-227.

COMMENT 2.7: An asthma attack is the result of an exacerbation of the oxidative stress from reactive oxygen species (ROS), manifested more frequently during the night or early morning, by the appearance of paroxysms with specific symptoms. Aggravating effects such as inflammation, hyperactivity and obstruction of the respiratory tract should be discussed.

REPLY 2.7: As mentioned in Reply 2.6, asthma control is a different concept than acute exacerbations of asthma symptoms. But we have now reviewed other determinants that have been reported as risk factors for reduced asthma control and added refs 5-8. We also discuss shortly potential mechanisms for reduced asthma control, although we think that very extensive discussion on potential mechanisms is out of scope of this manuscript.

COMMENT 2.8: Not lastly, table 1, which is a key element of the presented research lacks important information in the caption (values in the parentheses are percentages?). No statistical indicator was presented. Neither significance. ORs or RRs should be considered. Statistical power was not presented.

REPLY 2.8: We have now edited Table 1 to show more clearly that the values are numbers in each category and those in parentheses are percentages. This Table gives descriptive information on the study population, so statistical testing was not performed. Statistical testing is relevant in Table 2, in which it is marked by showing the 95% confidence intervals.

COMMENT 2.9: L150 presents a simplified causal pathway, dampness and associated molds being a component of the asthma triggers. This should be discussed according to the elements presented before.

REPLY 2.9: The causal pathway presented is from a previous publication including a systematic review on indoor dampness and mold exposures and development of asthma. We refer to that article in the present manuscript and think that very detailed discussion of potential mechanisms is beyond the scope of this manuscript, but we list shortly potential mechanisms of effects on asthma control on P.6, 234-238.

COMMENT 2.10: Limitations should be discussed. The approach was solely based on questionnaires. No monitoring and measurements were performed.

REPLY 2.10: We are discussing the limitations of this article in a systematic way in the Validity of Results. Because this was based on a very large population-based epidemiological study it was unfortunately not feasible to carry out dampness or mold measurements. This is now discussed in the Validity of results section.

COMMENT 2.11: English language should be checked by reformulating ambiguous statements and correcting typos (e.g. L 148 “was prosed”).

REPLY 2.11: English language has now been checked (applying Canadian English) and typos have been corrected.

Reviewer 3 Report

In the manuscript submitted by Jaakkola et al, presents the potential relations between indicators of indoor dampness and mold exposures at home and asthma control among adult subjects with physician-diagnosed asthma.

In my opinion, the manuscript lacks of enough scientific quality to be published in this version in  International Journal of Environmental Research and Public Health. Manuscript needs major revision . In order to help to authors to improve a future version of the manuscript, some specific comments are listed below.

 Specific Comments:

1.     Lines 30-38: Introduction: As general view, the introduction is too short and it lacks of a consistent argumenta line leading to readers to clear view of what and why you carried out this research.  How does current manuscript contributes to the existing literature? Which is the research gap? The manuscript presented for review does not show any critical analysis. It is not known in which direction the authors are going.

This section has to be rewritten.

2.     There are no explicit research hypotheses in the manuscript.

3.     Lines 86-105: The results are presented in two tables. The commentary on the results is insufficient. Discuss the obtained results in more detail. I recommend a more advanced statistical analysis. The importance of the presented results in the context of human health should be emphasized more.

4.     Write specifically  “Those reporting exposure to residential dampness and/or molds were somewhat more often women,…”

 5.     Line 107-155: In Discussion Authors should discuss the results and how they can be interpreted in perspective of previous studies and of the working hypotheses. The findings and their implications should be discussed in the broadest context possible and limitations of the work highlighted. Future research directions may also be mentioned. The Discussion should be rewritten. Indicate (emphasize) the originality of the obtained results.

 6.     Line: 157-162: Conclusions. It should be well written and highlight the present study's significant findings. Specifically how and where these results can be used. Furthermore, please consider that the conclusion is intended to help the reader understand why your research should matter to them. A conclusion is not merely a summary of the main results but a synthesis of key points and where you recommend new areas for future research.

 7.     Line 181- 195: It is recommended that the manuscript be supplemented with additional literature. Only 6 literature items. Self-citations account for as much as 67%.

Author Response

COMMENT 3.1: In my opinion, the manuscript lacks of enough scientific quality to be published in this version in  International Journal of Environmental Research and Public Health. Manuscript needs major revision. In order to help to authors to improve a future version of the manuscript, some specific comments are listed below.

REPLY 3.1: We have now made a major revision to improve the quality of the manuscript, and re-submit it here to the International Journal of Environmental Research and Public Health.

  • We have expanded the introduction by presenting previous studies of risk factors of asthma control measured by ACT
  • We have added 8 new references
  • We apply “poor asthma control” as an additional health outcome
  • We have added a new table on results
  • We have expanded discussion

 Specific Comments:

COMMENT 3.2: 1.     Lines 30-38: Introduction: As general view, the introduction is too short and it lacks of a consistent argumenta line leading to readers to clear view of what and why you carried out this research.  How does current manuscript contributes to the existing literature? Which is the research gap? The manuscript presented for review does not show any critical analysis. It is not known in which direction the authors are going.

This section has to be rewritten.

REPLY 3.2: We have now revised the Introduction and provide a clear statement on the hypothesis underlying this study and the objective that this epidemiologic study aims to answer.

COMMENT 3.3: 2.     There are no explicit research hypotheses in the manuscript.

REPLY 3.3: We have added a clear statement on the hypothesis underlying this study and the objective that this epidemiologic study aims to answer. P.2, lines 54-57.

 COMMENT 3.4: 3.     Lines 86-105: The results are presented in two tables. The commentary on the results is insufficient. Discuss the obtained results in more detail. I recommend a more advanced statistical analysis. The importance of the presented results in the context of human health should be emphasized more.

REPLY 3.4: We have expanded the result section by adding Table 3 and added more text describing the results.

COMMENT 3.5: 4.     Write specifically  “Those reporting exposure to residential dampness and/or molds were somewhat more often women,…”

REPLY 3.5: Our results show such exposure more often among women, we have clarified this. The number of exposed and the corresponding percentages are given in Table 1.

Comments and Suggestions for Authors

 COMMENT 3.6: 5.     Line 107-155: In Discussion Authors should discuss the results and how they can be interpreted in perspective of previous studies and of the working hypotheses. The findings and their implications should be discussed in the broadest context possible and limitations of the work highlighted. Future research directions may also be mentioned. The Discussion should be rewritten. Indicate (emphasize) the originality of the obtained results.

REPLY 3.6: We have rewritten the Discussion and have added a perspective to previous studies. We also present the original hypothesis underlying this study.

 COMMENT 3.7: 6.     Line: 157-162: Conclusions. It should be well written and highlight the present study's significant findings. Specifically how and where these results can be used. Furthermore, please consider that the conclusion is intended to help the reader understand why your research should matter to them. A conclusion is not merely a summary of the main results but a synthesis of key points and where you recommend new areas for future research.

REPLY 3.7: We have rewritten the conclusions highlighting its most important findings. And we provide shortly a suggestion how these results could matter future research as well as public health and clinical practice.

COMMENT 3.8:  7.     Line 181- 195: It is recommended that the manuscript be supplemented with additional literature. Only 6 literature items. Self-citations account for as much as 67%.

REPLY 3.8: We have now conducted a new literature search and added several references as well as text related to them.

Round 2

Reviewer 2 Report

The authors have improved the previous form and all my comments have been addressed. Readability and structure are corresponding to the requirements. Results have been better presented based on relevant statistical indicators and the associated discussion is meaningful. 

The resulting revised version contributed to the body of knowledge on this major environmental health issue. 

Author Response

Reviewer 2 has already accepted our changes.

Reviewer 3 Report

The authors wrote a new better manuscript. The authors addressed most of the problems. The manuscript is suitable for publication.

Author Response

Reviewer 3 has already accepted our responses.